# Thinking Before Coding: WebUI-to-Code Driven by Layout Reasoning and Consistency Rewards

## Abstract

In recent years, Multimodal Large Language Models (MLLMs) have made substantial progress in visual understanding and language generation, offering new opportunities for automating front-end web development. The WebUI-to-Code task, translating webpage design mockups or screenshots directly into structured HTML, has emerged as a promising paradigm for intelligent front-end engineering. However, existing MLLMs often exhibit significant limitations when applied to real-world webpages with complex layouts and diverse visual styles, including code compilation failures, severe layout misalignments. A key reason for these issues lies in the lack of structured, human-like cognitive processes—namely, the "thinking first, then coding" paradigm commonly followed by human developers. To address this gap, we propose a reinforcement learning framework that explicitly enhances the model's reasoning ability prior to code generation. Specifically, we introduce a structured layout reasoning stage and design a three-stage reward mechanism to supervise (i) the quality of layout reasoning, (ii) the accuracy of the generated code, and (iii) the consistency between the reasoning and the code. This reward formulation is designed to provide strong positive feedback from the reasoning process to the code generation outcome. To rigorously evaluate our approach, we construct and manually curate a benchmark consisting of 1.8K real-world webpages spanning multiple levels of layout complexity and visual detail. Experimental results demonstrate that our reasoning-enhanced method significantly improves the performance of the baseline model and achieves results comparable to or even surpassing much larger MLLMs baselines in terms of compilation success rate, layout fidelity, and styling accuracy.

## 1 Introduction

In recent years, Multimodal Large Language Models (MLLMs) have achieved significant progress in both visual understanding and natural language generationPlaat et al. (2024); Chen et al. (2025b); Zhang et al. (2024); Bandyopadhyay et al. (2025); Wang (2025); Xu et al. (2025); Chen et al. (2025b), paving the way for transformative applications in automated front-end web development. Among these, translating webpage screenshots or UI mockups directly into structured HTML code (*i.e.,* WebUI-to-Code) has emerged as a promising paradigm for enhancing development efficiency and enabling intelligent front-end software engineering. Despite the encouraging results of state-of-the-art code generation models on synthetic or simplified web layoutsPix2code; Lee et al. (2023); Beltramelli (2018); Lee et al. (2023), their performance often degrades substantially when applied to real-world webpages that feature complex layouts, diverse components, and fine-grained visual styling. In such challenging settings, generated code frequently suffers from critical issues, including a high rate of compilation errors, severe layout misalignment, and missing or redundant UI elements, which significantly hinder the reliability and usability of current solutions.

In this work, we identify a fundamental limitation in existing approaches: most current models attempt to solve the WebUI-to-Code task as a direct black-box mapping from images to code, overlooking the structured cognitive process inherent in human front-end development. At its core, the WebUI-to-Code problem requires transforming high-dimensional visual inputs into structured and executable code. Human front-end engineers typically begin by perceiving and abstracting the visual

information presented in a UI design or screenshot, such as decomposing the overall page layout, planning the spatial configuration of elements, and recognizing the stylistic attributes of each component. However, mainstream code generation models lack this explicit intermediate reasoning, that is the "think first (layout reasoning), then act (code generation)" paradigm. As a result, these models often struggle to robustly capture the underlying structure and fine-grained visual details of complex webpages, leading to suboptimal and unreliable code generation quality.

To address the aforementioned challenges, we propose a novel approach that leverages Reinforcement Learning to enhance explicit reasoningLiu et al. (2025); Zhou et al. (2025), significantly improving the performance of MLLMs in front-end code generation tasks. The core of our algorithm is twofold: (i) we explicitly require the model to output intermediate reasoning steps, which include a high-level understanding of the overall webpage layout (within `<think></think>`) and a structured description of the layout information (within `<layout></layout>`). This reasoning serves as a blueprint to guide the subsequent code generation process (enclosed within `<answer></answer>`),; (ii) we introduce a three-stage reward mechanism, incorporating the accuracy of layout reasoning, the correctness of the generated code, and the consistency between the reasoning and the code. This composite reward function ensures the correctness of both the intermediate reasoning and the final generated code, while also providing significant positive reinforcement from the reasoning outcomes to the code generation process.

During the evaluation phase, we construct a real-world website dataset as the benchmark from existing benchmarkSi et al. (2024); Guo et al. (2024); Lin et al. (2025), manually filtering approximately 1,800 high-quality real-world front-end code generation tasks. The dataset is divided into three difficulty levels, sufficiently covering the challenges posed by complex layouts and fine-grained details. In summary, our key contributions are threefold:

- We propose the first reinforcement learning framework for front-end code generation, which integrates webpage layout reasoning, code rendering, and structured information extraction. To ensure both the correctness and consistency of reasoning and code, we design a tri-partite reward mechanism that effectively promotes strong positive feedback from reasoning to code generation.

- We construct the largest benchmark for real-world webpage code generation. The dataset is manually filtered and categorized by difficulty levels, covering diverse layouts and visual complexities. Experimental results demonstrate that this benchmark poses significant challenges to current mainstream MLLMs.

- We train with only 450 generated webpage UIs paired with corresponding front-end code. Combined with our proposed reinforcement learning framework, this leads to substantial improvements in Qwen2.5-VL-7B's code generation performance, achieving or even surpassing larger-scale multimodal models in terms of compilation success rate, layout fidelity, and style correctness.

## 2 METHODOLOGY

### 2.1 REINFORCEMENT LEARNING FRAMEWORK FOR WEB-TO-CODE

#### 2.1.1 LAYOUT REASONING VIA STRUCTURED PROMPTS

To enhance the model's interpretability and controllability during code generation, we explicitly introduce a **structured reasoning stage** before HTML generation. Inspired by Chain-of-Thought prompting in reasoning tasks, we design a staged prompt format that guides the model to first perform free-form reasoning (*i.e.,* enclosed in `<think></think>` tags), followed by layout summarization (*i.e.,* enclosed in `<layout></layout>` tags), and finally generate code (*i.e.,* enclosed in `<answer></answer>` tags).

The core principle behind layout reasoning is that the model is expected to partition the webpage into semantically coherent regions based on the functional continuity of content, rather than arbitrary visual blocks. For example, a horizontal section at the top containing a logo and navigation links is semantically identified as a `header`, while a centered grid of images and texts may be grouped as the `main` content. Such semantic partitioning mirrors the way frontend developers conceptualize

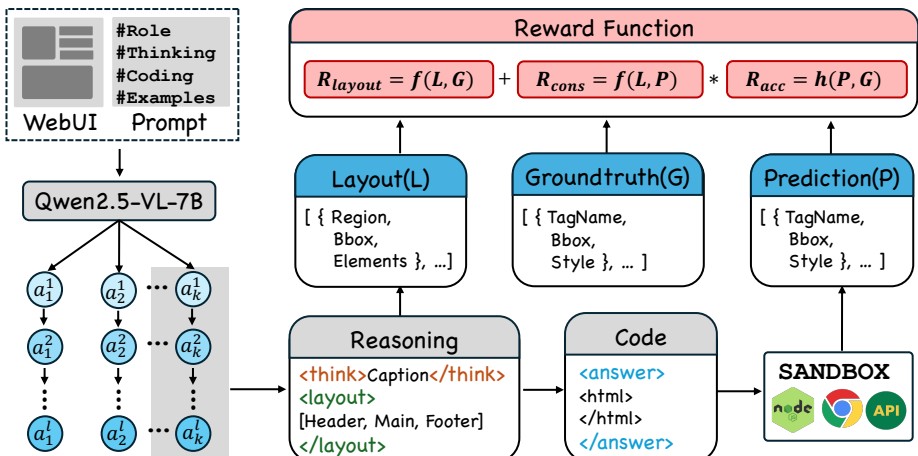

Figure 1: Reinforcement learning with GRPO for WebUI-to-Code, where composite rewards jointly optimize layout reasoning correctness, code reproduction accuracy, and the consistency between reasoning and generation

page structure. Once these regions are identified, the model predicts their spatial extent by returning bounding boxes $[x_1, y_1, x_2, y_2] \in [0,1]^4$, normalized with respect to the screenshot size. This process provides a coarse-to-fine scaffold for downstream code generation, grounding the placement of subsequent elements within their correct semantic zones. The details of prompt and output format are shown in Appendix A.1.

### 2.1.2 CODE-TO-STRUCTURE ABSTRACTION: EXTRACTING LAYOUT AND ELEMENTS

To enable effective fine-grained evaluation of the model's code generation outputs, we design a post-hoc abstraction module that converts the generated HTML + Tailwind code into structured representations. Specifically, we extract two parallel views: the layout view for spatial positioning, and the element view for visual styling. These abstractions are essential for comparing model predictions with ground-truth annotations and for defining reward signals in RL training.

**Layout-Level Abstraction.** At the layout level, we focus on the precise spatial positioning of each rendered visual element. To this end, we render the model-generated HTML using a headless browser and extract the bounding box for each DOM element. Each bounding box is represented as a 4-tuple $[x_1, y_1, x_2, y_2] \in [0,1]^4$, normalized by the page width and height. This allows us to evaluate the alignment and ordering of elements in the generated layout with respect to the ground truth.

**Element-Level Abstraction.** At the element level, we focus on semantic type and styling details. For each DOM node, we determine its HTML tag type (*e.g.,* `text`, `button`, `img`, `input`) and extract type-specific visual attributes. For instance:

- For `text` elements: we extract the `textual content`, `font-size`, `font-weight`, and `font-color`;
- For `button` elements: we extract the `textual content` and `background-color`;
- For `input` elements: we record the `border-radius`;
- For `image` elements: we fix the image source as `placeholder.jpg` and record image size;

The extracted layout and element information is merged into a unified JSON structure for each generated page. This structure forms the foundation for our downstream reward computation and metric-based evaluation. The example of the abstracted representation is shown in Appendix A.1.

### 2.1.3 REWARD FUNCTION DESIGN

To guide the model toward generating accurate, interpretable, and well-aligned HTML code from image inputs, we design a multi-component reward framework that supervises both the layout reasoning and code generation stages. Unlike traditional approaches that only reward the final code output, our reward formulation additionally incorporates the quality of the intermediate reasoning output, which we explicitly request from the model using `<layout>` tags.

The motivation behind this reward decomposition is on two-fold: (i) to ensure that both the reasoning process and the final answer are individually correct, and (ii) to encourage consistency between the reasoning and the code, such that the reasoning stage meaningfully contributes to the structure of the generated code. To this end, we decompose the reward into three components:

- **Code Accuracy Reward** ($R_{\text{acc}}$): measures the correctness of the final code in reproducing the layout and content of the input screenshot;
- **Layout Reasoning Reward** ($R_{\text{layout}}$): measures the quality of the predicted layout summary in the `<layout>` reasoning stage;
- **Consistency Reward** ($R_{\text{cons}}$): evaluates whether the layout structure described in the reasoning stage aligns with the structure manifested in the generated code.

**Code Accuracy Reward** ($R_{\text{acc}}$)   Given a predicted HTML document, we extract all rendered DOM elements using a headless browser and parse their tag type, bounding box, and style attributes as described in Sec. 2.1.2. We save the results in a predicted element set $\mathcal{E}_{\text{pred}} = \{e_1, e_2, ..., e_n\}$. Similarly, we extract a ground-truth element set $\mathcal{E}_{\text{gt}} = \{e_1^*, e_2^*, ..., e_m^*\}$ from the original webpage.

To compare these two sets, we first define a pairwise cost matrix $C \in \mathbb{R}^{m \times n}$ where each entry $C_{ij}$ represents the visual dissimilarity between ground-truth element $e_i^*$ and predicted element $e_j$:

$$C_{ij} = c(e_i^*, e_j) = \underbrace{c_{\text{type}}}_{\text{Tag mismatch}} + \underbrace{c_{\text{style}}}_{\text{Styling diff}} + \underbrace{c_{\text{bbox}}}_{\text{Position diff}} . \tag{1}$$

Each component is computed based on the type-specific attribute comparison rules introduced in Sec. 2.1.2. For example, $c_{\text{style}}$ for `text` elements considers font size, weight, and color differences; $c_{\text{bbox}}$ penalizes spatial displacement via normalized IoU. Given the cost matrix $C$, we apply the Hungarian algorithm **?** to find an optimal one-to-one assignment $\mathcal{M}$ between predicted and ground-truth elements:

$$\mathcal{M} = \text{Hungarian}(C), \quad \mathcal{M} \subseteq \mathcal{E}_{\text{gt}} \times \mathcal{E}_{\text{pred}}. \tag{2}$$

We then compute the code accuracy reward by transforming matched costs into similarity scores:

$$R_{\text{acc}} = \frac{1}{|\mathcal{E}_{\text{gt}}|} \sum_{(i,j) \in \mathcal{M}} \max(1 - C_{ij}, \, 0), \tag{3}$$

where higher reward corresponds to lower cost (*i.e.,* better match). A perfect alignment with zero cost leads to $R_{\text{acc}} = 1.0$, while high mismatches degrade the reward proportionally. This mechanism enables a smooth and differentiable reward surface for policy learning.

**Layout Reasoning Reward** ($R_{\text{layout}}$)   The layout reasoning reward focuses on evaluating how well the predicted regions align with the ground-truth regions in terms of both the types and counts of elements within each region. The first step is to filter out invalid or overlapping regions from the predicted layout. We define the set of valid regions $\mathcal{R}_{\text{valid}}$ as follows:

$$\mathcal{R}_{\text{valid}} = \{r \mid r \in \mathcal{R} \land \text{is\_valid}(r) \land \text{no\_overlap}(r)\} \tag{4}$$

Where $\mathcal{R}$ is the set of all predicted regions. is_valid($r$) ensures the region follows valid JSON format (i.e., contains bounding box and element details). no_overlap($r$) guarantees that the regions do not overlap, ensuring that each bounding box is unique and distinct from others.

For each valid region, we then map the `region_bbox` to groundtruth webpage and collect the set of element types( with their counts) in the spatial position of the area, denoted as $A_{\text{gt}}$. Let $A_{\text{layout}}$

represent the sets of predicted element types and counts described in `region_elements`. The similarity comparison between two sets is done using the Jaccard similarity:

$$\text{Jaccard}(A_{\text{layout}}, A_{\text{gt}}) = \frac{|A_{\text{layout}} \cap A_{\text{gt}}|}{|A_{\text{layout}} \cup A_{\text{gt}}|} \tag{5}$$

Once the Jaccard similarity is computed for each region, the layout reasoning reward $R_{\text{layout}}$ is calculated as the average similarity across all valid regions. Specifically:

$$R_{\text{layout}} = \frac{1}{N} \sum_{i=1}^{N} \text{Jaccard}(\text{gt}_i, \text{pred}_i), \tag{6}$$

where $N$ is the total number of valid regions, and $\text{gt}_i$ and $\text{pred}_i$ are the ground-truth and predicted element sets for the $i$-th region, respectively.

**Consistency Reward ($R_{\text{cons}}$)** While the layout reasoning reward ($R_{\text{layout}}$) encourages the model to produce a correct interpretation of the image, it does not guarantee that the generated code faithfully implements this interpretation. In practice, we observe that models may produce well-structured reasoning in the `<layout>` section but ignore it during actual code generation, leading to hallucinations or inconsistencies. To address this issue, we introduce the consistency reward $R_{\text{cons}}$, which explicitly enforces alignment between the predicted reasoning and the generated code.

The core idea is to measure whether the layout reasoning (*i.e.,* region summaries with element type counts) is consistent with the actual structure of the code output. That is, we compare:

- The set of regions and element counts extracted from the `<layout>` section (denoted as $\mathcal{R}_{\text{layout}}$), and
- The corresponding region-level element statistics extracted from the generated code (denoted as $\mathcal{R}_{\text{code}}$).

We extract all valid regions from the `<layout>` section using the same criteria as in $R_{\text{layout}}$: valid JSON format and no overlapping bounding boxes. For each region $r_i \in \mathcal{R}_{\text{layout}}$, we locate the corresponding region in the generated code by using its predicted bounding box and extract the actual rendered element types and counts (from DOM parsing). Let $A_{\text{layout}}$ and $A_{\text{code}}$ be the sets of element types and counts from reasoning and code, respectively. The consistency for that region is computed using Jaccard similarity:

$$r_{\text{cons},i} = \frac{|A_{\text{layout}} \cap A_{\text{code}}|}{|A_{\text{layout}} \cup A_{\text{code}}|} \tag{7}$$

The final consistency reward is the average Jaccard similarity across all valid regions:

$$R_{\text{cons}} = \frac{1}{|\mathcal{R}_{\text{layout}}|} \sum_{r_i \in \mathcal{R}_{\text{layout}}} r_{\text{cons},i} \tag{8}$$

**Final Reward Formulation.** To integrate the reasoning quality and execution accuracy into a unified reinforcement learning objective, we define the final reward as:

$$R = R_{\text{layout}} + R_{\text{cons}} \cdot R_{\text{acc}}$$

The multiplicative term $R_{\text{cons}} \cdot R_{\text{acc}}$ creates a gating mechanism: if the code is correct but inconsistent with the reasoning, the reward is penalized proportionally. This design prevents the model from ignoring the reasoning step and forces it to treat the `<layout>` prediction as a meaningful intermediate representation. In other words, we encourage the model to "do what it says"—only when the generated code aligns with the intermediate reasoning plan does it receive full credit.

Meanwhile, we include the additive term $r_{\text{layout}}$ to reward good reasoning in its own right, even if the code is imperfect. This ensures that the model maintains a clear understanding of visual structure and region composition, which is beneficial for generalization and interpretability. Overall, this reward formulation balances layout understanding and faithful execution, guiding the model toward generating accurate and interpretable front-end code grounded in visual reasoning.

### 2.1.4 Optimization with GRPO

We adopt Guided Reinforcement Policy Optimization (GRPO) Yang (2025); Chen et al. (2025a); Huang et al. (2025) to refine the image-to-code model. Given policy parameters $\theta$, screenshot $s$, and generated code $a$, GRPO maintains a guide policy $\pi_g$ (frozen backbone), optimizing the KL-regularized objective:

$$\theta_{t+1} = \arg\max_{\theta} \mathbb{E}_{a \sim \pi_\theta} \left[ R(s, a) - \beta \, \mathrm{KL} \left( \pi_\theta(\cdot \mid s) \, \| \, \pi_g(\cdot \mid s) \right) \right], \tag{9}$$

where $R(s, a)$ is our designed reward function (Sec. 2.1.3), and $\beta$ controls the trust region size. The objective is optimized via importance-weighted policy gradients with the guide policy fixed as a stable reference.

## 3 Experiments

### 3.1 Benchmarking

In this work, we select real-world websites as the evaluation dataset to comprehensively assess MLLMs for front-end code generation. Compared to synthetic datasets used in prior works (*i.e.,* WebsightLaurençon et al. (2024)), real-world websites exhibit significantly richer layout structures, greater variety in WebUI elements, and more intricate content details. These characteristics present substantial challenges to a model's capabilities in visual understanding, structural reasoning, and code synthesis, thus enabling a more rigorous evaluation of model performance under realistic and complex conditions.

According to an exhaustive review and analysis of existing benchmarks, we identify three major benchmarks containing front-end code generation tasks based on real websites: **Design2Code**Laurençon et al. (2024), **IWBench**Guo et al. (2024), and **WebUIBench**Lin et al. (2025). However, during our inspection, we observed that these benchmarks contain noisy samples that either fail to load completely, include intrusive pop-ups or advertisements, or are otherwise inaccessible.

To address this issue and ensure the quality of the evaluation dataset, we conducted a rigorous manual filtering and curation process on these benchmarks. Specifically, we excluded webpages with the aforementioned issues and carefully verified the integrity and completeness of each remaining webpage. Ultimately, we constructed a refined, high-quality evaluation dataset consisting of 1.8K real-world webpages.

### 3.2 Experiments Settings

**Baseline.** Our baseline model is Qwen2.5-VL-7Wang et al. (2024)B. To assess the effectiveness of our approach, we compare the reinforcement learning fine-tuned Qwen2.5-VL-7B, using GRPO, against several state-of-open-source and close-source MLLMs, including Qwen2.5-VL-32B, Qwen2.5-VL-72B and GPT-4oHurst et al. (2024), evaluated on our curated benchmark for front-end code generation tasks. During testing, we compared the above models using three types of prompts: **Direct Inference**, step-by-step thinking(**CoT**), and our designed structured layout-reasoning prompt(**Layout CoT**). The detailed prompt contents are provided in Appendix A.1.

**Training Details.** The RL training dataset consists of 450 high-quality, diverse examples, synthesized by a large pretrained model and rigorously filtered by human experts. These examples cover a wide range of canonical front-end code generation scenarios. We perform end-to-end reinforcement learning on Qwen2.5-VL-7B using the GRPO algorithm within the ms-swift framework, distributed across 8 NVIDIA A800 GPUs (80 GB each). Both input and generated sequence lengths are capped at 8,192 tokens. The training runs for 5 epochs, with a per-GPU batch size of 1 for both training and evaluation. We use an initial learning rate of $1 \times 10^{-6}$ and a sampling temperature of 0.9. To optimize distributed training efficiency, DeepSpeed ZeRO-3 is employed. During GRPO optimization, we generate seven rollouts per prompt and set the trade-off coefficient $\beta$ to 0.001.

| Model | CER.($\downarrow$) | Reward Score($\uparrow$) | | | | Visual Similarity($\uparrow$) | | | |
|---|---|---|---|---|---|---|---|---|---|
| | | Easy | Medium | Hard | Avg. | Easy | Medium | Hard | Avg. |
| GPT-4o | | | | | | | | | |
|   + Direct Inference($\dagger$) | 6.1 | 46.9 | 37.1 | 23.7 | 36.2 | 79.6 | 82.9 | 58.1 | 73.5 |
|   + CoT($\diamond$) | 3.2 | 46.9 | 38.3 | 30.7 | 38.6 | 80.8 | 84.5 | 84.6 | 83.3 |
|   + Layout CoT($\star$) | 1.4 | 50.5 | 42.8 | 35.4 | 42.9 | 85.2 | 84.8 | 84.1 | 84.7 |
| Qwen2.5-VL-7B | | | | | | | | | |
|   + Direct Inference($\dagger$) | 4.5 | 43.2 | 34.6 | 27.2 | 34.9 | 77.7 | 76.1 | 73.7 | 76.2 |
|   + CoT($\diamond$) | 7.6 | 42.7 | 33.6 | 28.0 | 34.5 | 66.9 | 66.1 | 66.0 | 66.6 |
|   + Layout CoT($\star$) | 1.9 | 45.9 | 35.5 | 29.7 | 36.7 | 80.9 | 80.6 | 80.0 | 80.8 |
| Qwen2.5-VL-32B | | | | | | | | | |
|   + Direct Inference($\dagger$) | 1.8 | 51.0 | 43.7 | 35.9 | 43.6 | 83.6 | 80.8 | 79.3 | 81.4 |
|   + CoT($\diamond$) | 1.4 | 51.3 | 44.1 | 35.7 | 43.8 | 85.1 | 85.9 | 86.1 | 86.0 |
|   + Layout CoT($\star$) | 0.5 | 50.6 | 41.2 | 31.8 | 41.2 | 85.6 | 86.3 | 85.0 | 86.1 |
| Qwen2.5-VL-72B | | | | | | | | | |
|   + Direct Inference($\dagger$) | 10.5 | 50.3 | 40.4 | 31.9 | 40.8 | 73.9 | 69.6 | 61.3 | 68.8 |
|   + CoT($\diamond$) | 8.4 | 47.1 | 38.0 | 30.8 | 38.5 | 80.4 | 77.8 | 77.3 | 78.6 |
|   + Layout CoT($\star$) | 0.4 | 46.3 | 33.0 | 23.2 | 33.9 | 84.9 | 83.7 | 82.5 | 84.0 |
| **Qwen2.5-VL-7B-RL(Ours)** | 0.9 | 50.1 | 42.3 | 36.1 | 42.6 | 83.9 | 84.7 | 84.3 | 84.7 |
| $\Delta$ (*vs* Qwen2.5-VL-7B) | +0.7$^\star$ | +4.2$^\star$ | +6.8$^\star$ | +6.4$^\star$ | +5.9$^\star$ | +3.0$^\star$ | +4.1$^\star$ | +4.3$^\star$ | +3.9$^\star$ |
| $\Delta$ (*vs* Qwen2.5-VL-32B) | -0.4$^\star$ | -1.2$^\diamond$ | -1.8$^\diamond$ | +0.2$^\diamond$ | -1.2$^\diamond$ | -1.7$^\star$ | -1.7$^\star$ | -1.8$^\diamond$ | -1.4$^\diamond$ |
| $\Delta$ (*vs* Qwen2.5-VL-72B) | -0.5$^\star$ | -0.2$^\dagger$ | +1.9$^\dagger$ | +4.2$^\dagger$ | +1.8$^\dagger$ | -1.0$^\star$ | +1.0$^\star$ | +1.8$^\star$ | +0.7$^\star$ |
| $\Delta$ (*vs* GPT-4o) | +0.5$^\star$ | -0.4$^\star$ | -0.5$^\star$ | +0.7$^\star$ | -0.3$^\star$ | -1.3$^\star$ | -0.1$^\star$ | +0.2$^\star$ | +0.0$^\star$ |

Table 1: Evaluation results under three metrics: code compilation error rate(CER.), rule-based reward function score, and visual similarity to webpage screenshots. The $\Delta$ indicate the best-performing result among Direct Inference, CoT, and Layout CoT settings.

| Model | Easy | | Medium | | Hard | | Average | |
|---|---|---|---|---|---|---|---|---|
| | Style | Layout | Style | Layout | Style | Layout | Style | Layout |
| GPT-4o | 57.5 | 2.7 | 44.8 | 3.3 | 34.1 | 2.4 | 45.4 | 2.9 |
| Qwen2.5-VL-7B* | 56.2 | 2.6 | 45.2 | 3.1 | 36.8 | 2.5 | 45.8 | 2.8 |
| Qwen2.5-VL-32B* | **61.1** | **3.2** | **52.3** | **3.7** | **42.6** | 2.7 | **52.1** | **3.3** |
| Qwen2.5-VL-72B* | 60.3 | 1.4 | 48.4 | 1.9 | 38.5 | 1.3 | 48.9 | 1.6 |
| **Qwen2.5-VL-7B-RL(Ours)** | 59.7 | 2.6 | 50.2 | 3.6 | **42.9** | **3.0** | 50.8 | 3.1 |

Table 2: Fine-grained evaluation of element styling and layout accuracy based on the rule-based reward decomposition. The * denote the best results among Direct Inference, CoT, and Layout CoT configurations reported in Tab 1. **Bold** entries represent the best performance in each category and the underline entries represent the second-best performance.

## 3.3 MAIN RESULTS

### 3.3.1 COMPILATION SUCCESS RATE IMPROVED SIGNIFICANTLY

Compared to prompting models to directly generate front-end code, instructing them to 'think first, then generate' led to lower compilation error rates for both the 32B and 72B models (*e.g.,* 32B dropped from 1.8% to 1.4%). However, the 7B model exhibited the opposite trend, with its error rate increasing from 4.5% to 7.6%. Upon closer inspection, we found that the 7B model tends to produce verbose and redundant reasoning, resulting in truncated HTML outputs or repetitive generation.

To address this, we proposed Layout Chain-of-Thought (Layout CoT) prompting. All models demonstrated significant reductions in compilation error rates under this approach (*e.g.,* 72B dropped from 10.5% to 0.4%). The reason is that Layout CoT encourages the model to reason about the number and spatial distribution of elements, thereby producing clearer layout structures and reducing the likelihood of truncation or repetition.

Building upon this insight, we applied a reward function based on layout reasoning to train Qwen2.5-VL-7B using reinforcement learning. This reduced its compilation error rate from 1.9% to 0.9%. The RL framework effectively guided the model to generate more compilable code, substantially improving the validity and executability of the outputs.

### 3.3.2 KEY PERFORMANCE IMPROVEMENTS

**Capability on Complex Webpages:** Our training strategy significantly enhanced the model's ability to handle complex webpages. As shown in Tab 1, comparing the performance of Qwen2.5-VL-7B under various prompts with the RL-enhanced version, we observe that performance gains were more prominent as webpage complexity increased. For instance, visual similarity improved from +3.0% on easy webpages to +4.3% on hard ones. This confirms that our reward design and RL strategy are particularly effective under high-complexity scenarios, enabling the model to learn more robust policies for complex layouts.

**Outperforming Larger Parameter Models:** The RL-trained Qwen2.5-VL-7B model approached or even outperformed larger parameter models. As shown in Tab 1, the RL-trained model's rule-based reward score lagged behind 32B by only 1.2% while outperforming 72B by 1.8%. In terms of visual similarity, our model reached 84.7%, slightly below 32B (86.0%) but well above 72B (78.6%). According to Tab 2, on the most complex webpages, our model even surpassed 32B by 0.3% in element attribute recognition and matched 32B in layout understanding (both at 2.7%). These results demonstrate the effectiveness of our efficient RL training scheme, enabling a smaller model to match or exceed the performance of larger models in front-end code generation, producing webpages with highly accurate layout and element understanding.

### 3.3.3 MORE OBSERVATIONS

**Increasing model size does not guarantee better performance on specific tasks.** As shown in Tab 1, the average layout understanding score for 72B is only 1.6%, significantly lower than the 7B model's 2.8%. This is further reflected in Tab 2, where prompting 72B to "think in layout then generate code" leads to a drop in rule-based reward from 40.8% to 33.9%, underperforming 7B (36.7%) and far behind 32B (41.2%). We attribute this to the 72B model's weaker ability in layout understanding, which substantially limits its performance on front-end code generation tasks.

Interestingly, applying Layout CoT to large models has opposing effects on visual similarity and rule-based rewards. For example, in Tab 1, 32B's rule-based reward drops from 43.6% to 41.2%, while visual similarity improves from 81.4% to 86.1%. The same trend is observed in 72B. We hypothesize that Layout CoT prompts promote coarse-grained understanding of the overall webpage structure, improving global layout quality but potentially distracting the model from fine-grained details.However, for smaller models with limited visual understanding capacity, Layout CoT helps them focus on a broader range of elements, leading to simultaneous improvements in both visual similarity (+4.6%) and rule-based reward (+1.8%).

## 4 ANALYSIS

### 4.1 EFFECT OF REWARD STRUCTURE ON TRAINING BEHAVIOR

To investigate how different reward formulations affect model learning dynamics, we conduct a controlled comparison across three reward configurations:

- $Reward_A = R_{\mathrm{cons}} \cdot R_{\mathrm{acc}} + R_{\mathrm{layout}}$, where accuracy and consistency are jointly optimized, while layout reasoning is independently encouraged.

- $Reward_B = R_{\mathrm{cons}} \cdot (R_{\mathrm{acc}} + R_{\mathrm{layout}})$, which applies consistency as a gating term to all reward signals.

- $Reward_C = R_{\mathrm{acc}} + R_{\mathrm{cons}} + R_{\mathrm{layout}}$, where code accuracy is optimized independently, and consistency affects only layout supervision.

Fig 2 depicts the evolution of the three sub-rewards $R_{\mathrm{acc}}$, $R_{\mathrm{layout}}$ and $R_{\mathrm{cons}}$ under the three reward formulations. A closer inspection of the curves allows us to draw several complementary insights beyond the high-level observations.

$Reward_A$**: Accuracy-driven yet stable.** As evidenced by the red curves in Fig 2 (a) and (b), both $R_{\mathrm{acc}}$ and $R_{\mathrm{layout}}$ reach the highest reward values at the end of training, indicating that this reward formulation effectively improves layout reasoning and the quality of the generated code.

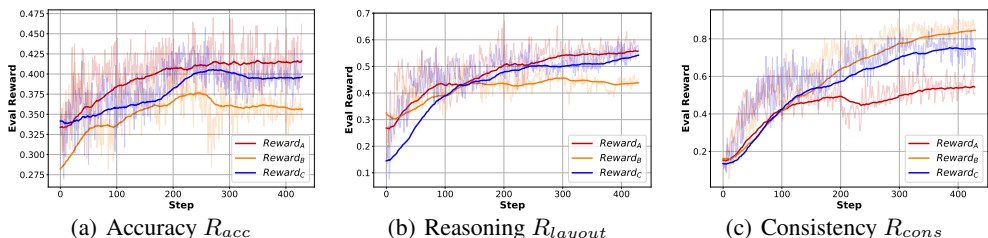

(a) Accuracy $R_{acc}$    (b) Reasoning $R_{layout}$    (c) Consistency $R_{cons}$

Figure 2: Trends of sub-reward functions during reinforcement training under $Reward_A$, $Reward_B$ and $Reward_C$ settings.

Because $R_{acc}$ is gated by $R_{cons}$, only samples that are both code-correct and reasoning-consistent can yield a large return. The $R_{layout}$, however, supplies a dense signal early on, preventing training stagnation due to reward sparsity. Consequently, the policy first learns to produce reliable layout plans and then, in order to earn $R_{acc}$, learns to "execute what it reasons," progressively refining code accuracy. This mechanism markedly strengthens the positive feedback from layout reasoning to code generation. Meanwhile, the same learning dynamics constrain the growth of $R_{cons}$ to a moderate, smooth trajectory, yet it still converges to values above 0.5, sufficient to maintain the desired alignment between reasoning and code throughout training.

$Reward_B$: **Consistency-centred but conservative.** The yellow curve in Fig 2 yields the highest $R_{cons}$; however, both $R_{layout}$ and $R_{acc}$ converge more slowly and reach the lowest final scores. Under this reward formulation, $R_{cons}$ functions as a strict gate that couples reasoning and code execution. The policy is therefore forced to improve consistency first; when the early–stage reasoning is immature, the reward remains zero for an extended period, causing a large proportion of rollouts to be rejected and leaving GRPO with almost no informative samples to update on. This severe reward sparsity during the initial phase hinders the optimisation of code accuracy, and although consistency rises quickly, neither layout reasoning nor code quality improves substantially.

$Reward_C$: **Ungated accuracy with weakened feedback.** It is noteworthy that, under this reward formulation, all three sub-rewards are obtained independently and impose no mutual constraints. As Fig 2illustrates, the blue curve for $R_{acc}$ rises during early training but subsequently declines, indicating that the code produced at the beginning can earn reward on its own; however, in the absence of a consistency gate, the model's generalisation ability stagnates and ultimately overfits the training samples. Meanwhile, $R_{layout}$ attains a level comparable to that of $Reward_A$, confirming that the model does learn an appropriate layout reasoning strategy. Nevertheless, because the generated code is decoupled from the reasoning process, the consistency signal fails to feed back into the final code generation stage, leaving overall code quality unimproved despite correct reasoning.

## 5    CONCLUSION

In this work, we investigate the reasoning deficiencies of MLLMs in the WebUI-to-Code task and propose a reinforcement learning framework that explicitly incorporates layout reasoning prior to code generation. By structuring the generation process into intermediate reasoning, layout planning, and final code synthesis, our method encourages models to "think before coding". We further introduce a tri-component reward design that jointly supervises layout reasoning accuracy, code correctness, and reasoning–code consistency. Experimental results on a curated dataset of 1,800 real-world webpages demonstrate that our approach significantly improves baseline performance, and achieves results comparable to or exceeding those of much larger MLLMs.

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
