# A  APPENDIX

## A.1  RELATED WORK

### A.1.1  MULTIMODAL REASONING

With the evolution of multimodal large language models(MLLMs), the chain-of-thought(CoT) reasoning mechanism has been extended to the multimodal domain, giving rise to multimodal chain-of-thought(MCoT). MCoT enhances the reasoning capabilities of MLLMs by simulating human-like step-by-step reasonging processes, significantly improving model performance in cross-modal complex tasks.MCoT has been widely adopteded in critical domains such as autonomous driving, embodied AI, robotics, and code generation, serving as a foundational technnology for achieving multimodal artificial general intelligence.

To investigate the robustness of MCoT in multimodal contexts, researchers have employed CoT promptingGao et al. (2024); Mitra et al. (2024); Wu et al. (2025) and constructed datasets with step-level reasoning annotationsThawakar et al. (2025); Xu et al. (2024); Zhang et al. (2024); Shao et al. (2024), followed by supervised fine-tuning to enhance MLLM reasoning capabilitiesXu et al. (2024); Yao et al. (2024); Thawakar et al. (2025); Shao et al. (2024); Cheng et al. (2024). Additionally, inspired by the success of DeepSeek, recent studies have leveraged reinforcement learning algorithms (e.g., Group Relative Policy Optimization, GRPO) to enable MLLMs to self-improve their reasoning through reward signalsZhang et al. (2024); Dong et al. (2024); Yang (2025); Chen et al. (2025); Wang et al. (2024); Huang et al. (2025); Liu et al. (2025); Zhou et al. (2025). These methods not only strengthen structured generation but also introduce novel paradigms for intermediate state modeling in complex tasks.

### A.1.2  CONVERTING THE WEBPAGE TO HTML CODE

Converting the webpage to HTML code, a crutial aspect of front-end automation, requires the integration of multiple capabilities such as image understanding, visual layout parsing and structured code generation. This task serves as a key benchmark for evaluating MLLMs' multimodal reasoning proficiency.

To advance this task, WebSightLaurençon et al. (2024) pioneered the construction of a large-scale synthetic dataset to train models for end-to-end webpage generation.However, the limited diversity of synthetic data constrained generalization performance. Subsequent efforts shifted focus to evaluation frameworks. For instance, Design2CodeSi et al. (2024) curated real-world webpages as a benchmark, revealing MLLMs' deficiencies in layout comprehension and element recognition. IWBenchGuo et al. (2024) further introduced evaluation metrics such as Element Accuracy and Layout Accuracy and proposed a 5-step MCoT prompting chain, significantly improving models' structural understanding and generation precision. Web2CodeYun et al. (2024) leveraged more capable MLLMs to refine existing datasets, constructing an instruction-following dataset and introducing an evaluation framework that combines structural question-answering with code generation, thereby enhancing semantic comprehension. WebUIBenchLin et al. (2025) is a benchmark dataset for front-end code generation on real-world webpages. It further proposes sub-dimension evaluations of MLLMs' code generation capabilities and provides high-quality webpage data across five categories.

Inspired by the research above, we explored the integration of multimodal reasoning with front-end code generation. Our approach leverages reinforcement learning to significantly enhance the model's capability in generating complex webpage code according to the given screenshot.

## A.2  PROMPT AND OUTPUT FORMAT

In the `<layout>` stage, we explicitly instruct the model to summarize the semantic structure of the webpage by returning a structured list of layout regions. Each region must follow a fixed JSON-like schema:

```
{
  "region_name": "main",
  "region_bbox": [x1, y1, x2, y2],
```

```
  "region_elements": { tagType: Numbers }
  // tagType can be one of ['text','input','button','image','block']
}
```

An example of the abstracted representation from HTML code is shown below:

```
{
  "tagType": "text",
  "text": "Welcome to WebApp",
  "font": "24px, bold, rgb(34,34,34)",
  "bbox_2d": [0.12, 0.08, 0.86, 0.15]
}
```

Prompt for direct answering.

```
# Role
You are a frontend development assistant who has just received a webpage
screenshot. Please produce the corresponding HTML and Tailwind CSS code.
All blue blocks in the screenshot represent image elements. Please use
the  tag or background-image style to implement them, and use rick.
jpg as default image path.

# Coding Rules
Return a single HTML document encapsulated within <html></html> tags. Do
not include any JavaScript, external files, or external links. Do not
include markdown "```" or "```html" at the start or end. Please return
only the code in the following JSON format: {"code":"<html></html>"}
```

Prompt for thinking step by step.

```
# Role
You are a frontend development assistant who has just received a webpage
screenshot.Please think step by step and produce the corresponding HTML +
 Tailwind CSS code. All blue blocks in the screenshot represent image
elements. Please use the  tag or background-image style to implement
 them, and use rick.jpg as the default image path.

# Coding Rules
Return HTML document encapsulated within <html></html> tags. Do not
include any JavaScript, external files, or external links. Do not include
 markdown "```" or "```html" at the start or end. The output must follow
this exact structure (case, order, and wrapping tags unchanged):

# Output Examples
<think>
Step1:
Step2:
</think>
<html>
   <!-- your HTML + Tailwind here -->>
</html>
```

Prompt for layout reasoning.

```
# Role
You are a step by step frontend engineer expert proficient in HTML/CSS
and Tailwind. Your task is to analyze a webpage screenshot and generate
accurate HTML/CSS code. Please use Tailwind CSS for styling and implement
 webpage layout.

# Thinking Principles
- Identify the main structural components and layout of the webpage (such
 as header, nav, main, footer, and sidebar).
- Within a specified layout region, locate the positions and scan all
recognizable element slots(such as text, input, button, img and block,
where block is the areas with a background color that is not white).
```

```
-  Summarize the layout regions with the bounding box of position and tag
type,numbers within the region in the following format:
[{
"region_name" : "main",
"region_bbox": [x1,y1,x2,y2],
// x1,y1,x2,y2 are all floats, x2 > x1, y2 > y1
"region_elements": { tagName : Numbers}
// tagName can be one of ["text","input","button","img","block"]
}]
-  Return the list set of ** layout Summarization ** enclosed in <layout
></layout> tags.

# Coding Rlues
-  Do not use any other comment syntax.
-  All blue blocks in the screenshot represent image placeholder. Please
use rick.jpg as the default image path.

# Output Examples
<think>Your current reasoning.</think>
<layout>[{"region_name":"",...},{"region_name":"",...}]</layout>
<answer>
 <!DOCTYPE html><html lang="en"></html>
</answer>
```

## A.3 EVALTION DATASET CONSTRUCTION

To quantify the layout complexity of web page screenshots or images, we propose a composite scoring algorithm based on three equally weighted metrics: element count, type diversity, and spatial coverage. For each image, all elements with their bounding boxes and tag types are extracted. The three metrics are computed as follows: (1) element count score: normalized by dividing the total number of elements by 200 and capped at 1; (2) type diversity score: normalized by dividing the number of distinct tag types by 5; and (3) union area score: the total area covered by all bounding boxes, calculated using a sweep-line algorithm to account for overlaps.

The final complexity score is obtained by summing the three metrics with equal weights. Images are then categorized into three difficulty levels based on scores: Easy (lowest 25%), Medium (middle 50%), and Hard (highest 25%). Out-of-bound or invalid bounding boxes are excluded to ensure robust computation. This procedure provides a quantitative measure of layout complexity suitable for dataset stratification and difficulty-aware analysis.

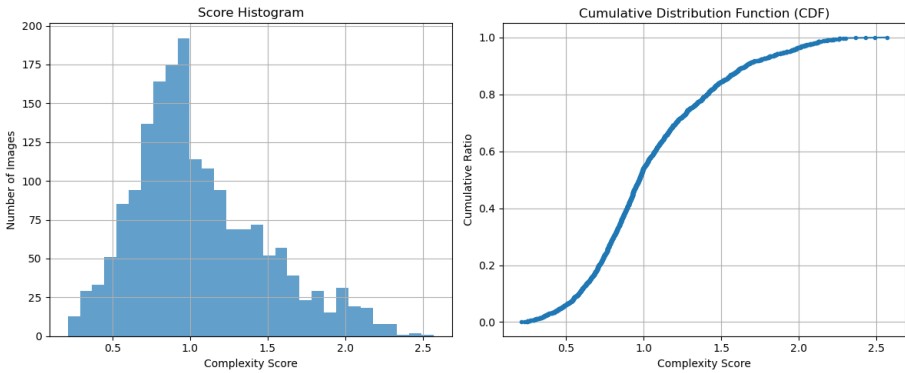

Figure 1: Distribution of Image Complexity Scores via Histogram and CDF. It illustrates the number of images across different complexity score intervals (left) and the cumulative proportion of images with scores up to each interval (right).

As shown in Fig 1, the score histogram on the left and the Cumulative Distribution Function (CDF) chart on the right in the figure jointly verify the effectiveness of the image complexity scoring al-

gorithm. In terms of distribution characteristics, the histogram presents a reasonable right-skewed pattern, which conforms to the natural data rule that "simple samples are the majority while complex samples are scarce". The CDF chart on the right provides further quantitative verification: the score ranges of the easy/mid/hard three-level samples are highly consistent with the complexity classification standards defined by the algorithm. Moreover, the 50th percentile (approximately 1.0) indicates that the overall complexity of the dataset is moderate, with no extremely abnormal scores. In conclusion, the data distribution in the charts is fully consistent with the design logic of the algorithm, which proves that the scoring algorithm can effectively distinguish differences in image complexity and provide a reliable basis for subsequent sample classification and model evaluation.

## B   LLM USAGE

In the preparation of this manuscript, LLM was employed solely as a writing aid to improve clarity, readability, and linguistic expression. Its use was limited to tasks such as translation, grammar correction, and language polishing.

The LLM was not used for generating any of the scientific content of the paper, including the formulation of research ideas, method design, experimental planning, data analysis, or interpretation of results. All intellectual contributions and research decisions were made solely by the authors.

The authors take full responsibility for the accuracy, integrity, and originality of the content presented in this manuscript and affirm that the use of the LLM did not compromise these standards.