# OpenReview forum: "Thinking Before Coding: WebUI-to-Code Driven by Layout Reasoning and Consistency Rewards"
_ICLR.cc/2026/Conference — ICLR 2026 Conference Withdrawn Submission_

### Official Review · Reviewer_Wy2m · 2025-10-28

**Soundness:** 2
**Presentation:** 1
**Contribution:** 2
**Rating:** 2
**Confidence:** 3

**Summary:**

The paper proposes a training recipe for UI-to-Code tasks by introducing intermediate layout representations (wrapped in `<layout>` and `</layout>`) in addition to CoT and code outputs.

The paper introduces a tri-component reward which supervises (i) code accuracy, (ii) layout reasoning accuracy, and (iii) reasoning–code consistency. The reported results from 1.8K-page real-world benchmark (from Design2Code, IWBench, WebUIBench) shows performance gains of Qwen2.5-VL-7B finetuned with RL on the proposed rewards compared with base model baselines.

**Strengths:**

- Novel consistency-gated tri-reward design that explicitly ties reasoning fidelity to code execution; structured `<think>/<layout>/<answer>` design.

- Structured parsing and comparisons of HTML DOM elements, which could be a more reliable and generalizable metrics than some of the existing metrics in prior benchmarks.

**Weaknesses:**

- The paper has reproducibility issues. The authors suggested that prompting details, layout representations, and experiment details will be included in Appendix, but the Appendix is missing. There's no definitions or algorithms for visual similarity or the reward cost matrices. Evaluation dataset curation and quality-filtering are under documented.

- The authors proposed a layout reasoning method by introducing `<layout>`, `</layout>` between the think and answer tags. However, the effectiveness of such approach is under justified: Table 1 shows Layout-CoT has mixed effects on model performance with no clear gains; and the authors have noted that the model tend to ignore the `<layout>` section during code generation, and hence the need for `R_consistency`. Without careful ablation studies against RL training with end-to-end generation, it is unclear whether the proposed layout reasoning module is beneficial or a distraction to the model's performance.

- Limited evaluation scope: Only GPT-4o and Qwen-2.5-VL family, omits other strong MLLMs.

- The paper asserts “Increasing model size does not guarantee better performance” and attributes 72B’s underperformance to weaker layout understanding, but offers no failure analysis or diagnostics to substantiate these hypotheses. Similarly, the discussion around Layout-CoT in section 3.3 remains speculative.

- The authors' claim on the curation of "the largest benchmark for real-world webpage code generation" is an overstatement. The 1.8k benchmark is a cleaned merge of three well-established benchmarks (Design2Code, IWBench, WebUIBench); transparency on inclusion/exclusion rules is limited.

- Presentation quality: Numerous typos and formatting glitches reduce clarity.

**Questions:**

- Please provide the missing appendix with: exact prompt templates and tag schema, full cost matrix formulas, training hyperparameters/seeds, and dataset curation criteria + licensing.

- Can the authors provide an ablation study on whether proposed `<layout>` stage is necessary for performance improvements?

- Please means + std over multiple seeds and significance test. Are the comparisons robust to sampling noise? Do the metric differences align with human preferences/judgments?

- Please broaden the evaluation to additional strong MLLMs/systems (and, if feasible, non-Tailwind targets such as React).

- Can the authors provide some analysis on common failure modes of Qwen-2.5 72B? When would larger models tend to fail while smaller models do not?

- Please proof read the manuscript and address typos & formatting issues.

---

### Official Review · Reviewer_H2HU · 2025-10-31

**Soundness:** 2
**Presentation:** 2
**Contribution:** 2
**Rating:** 2
**Confidence:** 4

**Summary:**

This work proposes thinking before coding for WebUI code generation, where authors use GRPO to train models to think about the layout, etc, before generating code, and design specific code accuracy reward, layout reasoning reward, and consistency reward. Evaluation on 1800 examples shows improvement over prompting baselines.

**Strengths:**

1. The authors construct a 1800 example benchmark for webpage code generation.
2. By training on 450 webpages, the authors improve the performance of Qwen2.5-VL_7B to be on par with larger models.

**Weaknesses:**

1. The thinking-before-coding seems already a common pattern for frontier models like GPT-5 and Claude, which limits the novelty and impact of this work.
2. The description and examples of the collected datasets are limited.
3. There seem to be a lot of detailed design choices in the reward, which are not fully justified.
4. How are the used metrics/rewards correlated with human annotators?

**Questions:**

See Weaknesses.

---

### Official Review · Reviewer_vQid · 2025-10-31

**Soundness:** 2
**Presentation:** 2
**Contribution:** 3
**Rating:** 2
**Confidence:** 4

**Summary:**

The authors propose a three-part reward function (based on layout reasoning, code correctness, and consistency between reasoning and generated code) to enhance MLLMs for front-end UI code generation. They fine-tune Qwen2.5-VL-7B-Instruct using GRPO on 450 training samples over 5 epochs. A new benchmark is introduced by filtering high-quality samples from Design3Code, IWBench, and WebUIBench, resulting in roughly 1,800 samples. The authors compare their trained model against other capacities of Qwen2.5-VL-Instruct models and GPT-4o using various prompting strategies.

**Strengths:**

- Framing the task completion process as a sequence of reasoning, layout generation, and then code generation is a great idea and aligns well with how MLLMs can structure complex outputs.
- The three-part reward design is interesting and well-motivated, especially the use of the consistency reward R_{cons} as a gating mechanism for the accuracy reward R_{acc}.

**Weaknesses:**

Although the idea is good, I believe the experimental section lacks sufficient baselines and prior benchmarks and important details, which limits transparency and the strength of the comparisons. Including more results from previous literature would provide better context. Here's a break down of the points:

- The related work section does not mention several relevant recent benchmarks (such as WebMMU [1]), which are important for situating the contribution.

- The baseline coverage is limited. Evaluating the model on Design2Code, IWBench, and WebUIBench would allow for fairer comparisons with both open- and closed-source models beyond Qwen2.5-VL-Instruct. Training on Qwen2.5-VL-3B and also including other model families like InternVL2.5, InternVL2, or Gemini in the comparisons would further strengthen the analysis.

- The description of the RL training dataset lacks transparency. The authors state that it includes “450 high-quality, diverse examples, synthesized by a large pretrained model and rigorously filtered by human experts. These examples cover a wide range of canonical front-end code generation scenarios,” but they do not specify which model was used or define what constitutes “high-quality” and “diverse” or "wide-range".

---

Minor Errors:
- \citep{} should be used instead of \citet{} in multiple places, e.g:
- Line 038 typo: generationPlaat et al. (2024);
- Line 061 typo: reasoningLiu et al.
- Line 073 benchmarkSi et al.
- Missing citations: Line 193 Hungarian algorithm ?
- Misleading colors in Tab 1, could have green for the "improvements" and red for the "decreases" in performance.
- In Table 1, the reported delta for CER between Qwen2.5-VL-7B-RL (0.9) and Qwen2.5-VL-7B (1.9) should be 1.0, not 0.7, indicating a larger improvement than reported.


[1] WebMMU: A Benchmark for Multimodal Multilingual Website Understanding and Code Generation

**Questions:**

Please refer to the weaknesses section.

---

### Official Review · Reviewer_Pv2q · 2025-11-02

**Soundness:** 3
**Presentation:** 3
**Contribution:** 3
**Rating:** 4
**Confidence:** 4

**Summary:**

This paper proposes an RL training method for the Design2Code task. The VLM is prompted to include free-form reasoning in <think></think> tags, layout summarization in <layout></layout> tags, and finally the generated code in <answer></answer> tags in the output.

The corresponding reward function has two parts:
1. The code accuracy reward computes the match between the DOM elements of the rendered webpage and the oracle webpage
2. The layout reasoning reward computes the similarity between the predicted layout regions and the oracle regions
3. The consistency reward computes the overlap between the layout section and the code section of the model output

The combined reward is then optimized through GRPO. By doing RL on Qwen2.5-VL-7B, the performance can exceed the non-finetuned models and even match GPT-4o.

**Strengths:**

- Strong empirical results on the Design2Code task.

- The method is quite straightforward and intuitive.

**Weaknesses:**

- I'm not quite satisfied with your ablations. Can you add some ablations on the importance of incorporating each reward component? What happens if you do not include the consistency reward? And what happens if you ignore the layout reasoning altogether and only include the actual code and compute the code accuracy reward? I think such ablations are important to justify your final reward design.

- It'll be nice to include some qualitative examples of model generations before and after RL to illustrate the benefit from doing DL.

**Questions:**

- Missing reference on line 193/193 ("Hungarian algorithm ?").

- The citation format is off for WrbSight (line 289/290)

- Wrong citation on 296/297 (Design2Code should be Si et al.?); also wrong citation formats in this line.

---

### Note · Authors · 2025-11-16

I have read and agree with the venue's withdrawal policy on behalf of myself and my co-authors.